

# Adaptive learning algorithm based price prediction model for auction lots—deep clustering based interval quoting

Da Ke[1], Xianhua Fan[2] and Muhammad Asif[3]

[1] School of Management, Huazhong University of Science and Technology, Wuhan, Hubei, China
[2] School of Economics and Management, China University of Geosciences, Wuhan, Hubei, China
[3] Department of Computer Science, National Textile University, Faisalabad, Punjab, Pakistan

## ABSTRACT

This article addresses the problem of interval pricing for auction items by constructing an auction item price prediction model based on an adaptive learning algorithm. Firstly, considering the confusing class characteristics of auction item prices, a dynamic inter-class distance adaptive learning model is developed to identify confusing classes by calculating the differences in prediction values across multiple classifiers for target domain samples. The difference in the predicted values of the target domain samples on multiple classifiers is used to calculate the classification distance, distinguish the confusing classes, and make the similar samples in the target domain more clustered. Secondly, a deep clustering algorithm is constructed, which integrates the temporal characteristics and numerical differences of auction item prices, using DTW-K-medoids based dynamic time warping (DTW) and fuzzy C-means (FCM) algorithms for fine clustering. Finally, the KF-LSTM auction item interval price prediction model is constructed using long short-term memory (LSTM) and dual clustering. Experimental results show that the proposed KF-LSTM model significantly improves the prediction accuracy of auction item prices during fluctuation periods, with an average accuracy rate of 90.23% and an average MAPE of only 5.41%. Additionally, under confidence levels of 80%, 85%, and 90%, the KF-LSTM model achieves an interval coverage rate of over 85% for actual auction item prices, significantly enhancing the accuracy of auction item price predictions. This experiment demonstrates the stability and accuracy of the proposed model when applied to different sets of auction items, providing a valuable reference for research in the auction item price prediction field.

# INTRODUCTION

The auction market is a complex and dynamic environment that spans various commodity classes, including art and antiques, real estate, and industrial equipment. Price formation in this market is influenced by numerous factors, such as the quality of the goods, their scarcity, market supply and demand, the psychological expectations of bidders, and their financial situation. This complexity makes price forecasting for auction items particularly

Corresponding author
Xianhua Fan, Fanxh@cug.edu.cn

challenging yet crucial. Accurate price range prediction is essential in the auction market. It helps auctioneers set more reasonable starting and reserve prices, maximizing auction efficiency and revenue. For bidders, understanding the potential price range of goods aids in developing more rational bidding strategies, avoiding blind bids and excessive competition.

Traditional methods of predicting auction item prices have achieved specific results; however, these methods often provide a fixed prediction value. In actual auctions, bidders need a price range based on their situation and the market environment. Fixed-value predictions do not meet the exact needs of bidders. Additionally, due to the complexity and dynamism of the auction market, traditional prediction models often struggle to adapt to rapid market changes. When the market environment shifts, the predictive accuracy of these models significantly decreases.

In recent years, deep learning technology has achieved remarkable results in image recognition and natural language processing (*Birkeland & AlSkaif, 2024*; *Tang et al., 2024*). Deep learning models possess powerful feature learning and representation capabilities, enabling them to handle complex nonlinear relationships and offer new auction item price prediction approaches. Adaptive learning algorithms can dynamically adjust model parameters based on historical data, adapting to market changes. In the context of auction item price prediction, adaptive learning algorithms can help models better capture market dynamics and improve prediction accuracy (*Nie et al., 2024*). Consequently, an auction price prediction model integrating deep learning and adaptive learning algorithms can more effectively capture market dynamics and enhance prediction precision. However, existing deep learning models require substantial amounts of labeled data for training, and in the realm of auction item price prediction, high-quality labeled data may be relatively scarce. This scarcity can result in insufficient model training and suboptimal prediction outcomes (*Wu et al., 2024*).

Moreover, the initial parameter settings often influence autonomous learning algorithms' performance. Inappropriate initial parameters can cause the algorithm to converge to a local optimal solution rather than a global one. Additionally, while autonomous learning algorithms can dynamically adjust model parameters based on historical data, their adaptability may still be limited when confronted with the rapidly changing environment of the auction market.

Therefore, to address these challenges, this article employs a deep clustering algorithm to classify auction items, aiming to uncover price patterns and characteristics across different categories of commodities. It integrates an adaptive learning algorithm to develop an auction item price prediction model capable of forecasting price intervals for these commodities. The specific contributions of this study are outlined as follows:

1. Dynamic class spacing adaptive learning: This approach handles confusing classes by computing classification distances based on differences in predicted values from multiple classifiers within the target domain. Identifying and segregating confusing classes enhances the clustering of similar samples in the target domain and widens class distances. This improves the generalization capability of the source domain model and enhances classification accuracy on the target domain.

2. Dual clustering algorithm: A dual clustering algorithm is constructed to achieve fine clustering of auction items. Considering both the temporal characteristics and numerical variability of auction item prices, this approach utilizes the K-medoids clustering algorithm based on dynamic time warping (DTW) and fuzzy C-means algorithms. It incrementally clusters dynamic characteristics and numerical values to provide detailed insights into price dynamics.

3. KF-LSTM deep learning model: The article introduces the KF-LSTM model based on double clustering. This model leverages long short-term memory (LSTM) networks and integrates the results of dual clustering for deep learning prediction. Each class cluster obtained from dual clustering is separately trained using LSTM models, which are then utilized to predict output sequences of auction item prices based on date-linked predictive features.

## RELATED WORKS

### Domain adaptive learning algorithm

Domain adaptation (DA) learning algorithms (*Hu et al., 2024*) primarily explore strategies for mitigating domain bias between source and target data distributions. These approaches leverage similarities and discrepancies between domains to transfer and apply models trained in the source domain, thereby enhancing classification performance on the target domain. Recent research has focused on several key directions. One such direction involves employing the maximum mean discrepancy (MMD) (*Yu et al., 2024*) method to mitigate domain bias. The DeepAdaptationNetwork (DAN) introduced in literature (*Xu et al., 2024*) embeds task layer representations into a kernel Hilbert space, aligning mean embeddings across different domain distributions. Joint distribution adaptation (JDA), proposed in the literature (*Qian, Luo & Qin, 2024*), adapts source and target domain edge distributions and conditional distributions through dimensionality reduction, integrating them into an optimization objective. Additionally, the JointAdaptation Network (JAN), as proposed in the literature (*Liu, Peng & El-Latif, 2023*), extends DAN and JDA frameworks by aligning joint distributions of input features and output labels across domain-specific layers using the joint maximum mean discrepancy (JMMD) criterion. This approach integrates domain adaptation and adversarial learning in deep networks to maximize network JMMD, enhancing distinguishability between source and target domain distributions through adversarial training strategies.

Another research avenue in domain adaptive algorithms focuses on deep learning methods leveraging adversarial learning. Following the introduction of generative adversarial networks (GAN) in literature (*Chakraborty et al., 2024*), adversarial learning principles were incorporated into domain adaptive algorithms to address domain bias. However, traditional adversarial-based methods often underperform because they solely align source and target domains adversarially without deeply exploring deep data distribution disparities between them. Many current algorithms aim to tackle this limitation.

Another promising direction involves clustering-based pseudo-labeling methods. Introduced in *Guo, Yin & Yang (2024)*, this approach begins by clustering unlabeled

sample features from the target domain. Subsequently, pseudo-labels are generated based on these clusters and utilized for supervised training to optimize model performance on the target domain. This iterative process continues until convergence. While clustering-based pseudo-labeling methods can enhance pseudo-label quality through model optimization, they are susceptible to noise introduced by pseudo-labels. Insufficient generalization ability of pre-trained networks from the source domain contributes to this noise.

Moreover, challenges such as unknown target domain categories and limitations of clustering algorithms further exacerbate pseudo-labeling noise. *Yang, Shao & Yang (2023)* proposes training two identical networks concurrently, progressively capturing target domain data distribution and refining pseudo-labeling for improved network training. *Nguyen (2023)* introduces the joint application of classification and ternary loss in supervised training, while *Zoppi et al. (2023)* explores its application in unsupervised training scenarios.

## Deep clustering algorithm

Clustering is an essential algorithm in current data mining. Still, with the complexity of data, traditional clustering methods can no longer handle high-dimensional data types, so it is becoming increasingly crucial to downscale high-dimensional data using more powerful models. Since the essence of deep learning is to capture the excellent features of data by automatically extracting features through multi-layer neural networks, deep clustering has been proposed as joint optimal representation learning and clustering.

From the model design perspective, existing deep clustering algorithms are divided into two main categories: models based on traditional clustering ideas (*Gormley, Murphy & Raftery, 2023*) and neural networks (*Lazcano, Herrera & Monge, 2023*). These two main classes of methods have their own merits and aim to improve the accuracy and efficiency of clustering through different mechanisms. Clustering-based models are usually deep extensions or improvements of traditional clustering algorithms, such as K-means and spectral clustering in deep learning. For example, the K-means-based deep clustering method (*Bisen et al., 2023*) can significantly improve the performance of clustering compared to the traditional K-means algorithm by combining the feature extraction capability of deep learning. However, this method appears incompetent in dealing with data with non-convex clustering shapes. On the other hand, the deep clustering (*Peng et al., 2023*) method based on spectral clustering can handle non-convex-shaped data but still needs to improve performance.

Subspace-based clustering (*Jia et al., 2023*) attempts to leverage neural networks' powerful feature extraction capabilities, mainly showcasing its unique advantages when dealing with high-dimensional data. The auction market involves many high-dimensional data, such as bidders' historical behavior, bidding strategies, auction item attributes, market trends, *etc*. Subspace clustering methods can effectively handle these high-dimensional datasets, learning intrinsic structures like bidder behavior patterns and auction item value assessment models through feature extraction. This enhances the accuracy of auction outcome predictions and helps auction houses better understand market demands and

optimize auction strategies. However, scalability becomes challenging as these methods experience rapid increases in time and space complexity with larger datasets.

Alternatively, deep clustering methods based on probabilistic models like Gaussian mixture models (*Hamdi et al., 2023*) and information-theoretic approaches such as mutual information (*Wan et al., 2023*) offer diverse clustering strategies. These methods provide a wide range of clustering strategies. The model can learn the distribution characteristics of normal bidding behavior in auctions by performing clustering analysis on large amounts of historical data. Any behavior deviating from this distribution can be considered a potential risk, triggering further investigation and review. However, they often encounter high computational demands, slow convergence, and unstable training. Methods utilizing Kullback–Leibler divergence (*Golzari Oskouei, Balafar & Motamed, 2023*), while demanding in network complexity and training, may exhibit performance limitations due to their depth. Recently, deep clustering methods integrating generative adversarial networks and comparative learning (*Ros, Riad & Guillaume, 2024*) have gained traction for their flexibility and efficacy in clustering tasks, introducing novel learning mechanisms. However, challenges such as convergence issues with generative adversarial networks and handling positive and negative sample pairs in contrastive learning remain.

In conclusion, advancing deep clustering algorithms requires continual innovation beyond traditional clustering methods, harnessing neural networks' potent feature extraction capabilities, and balancing model design and algorithmic optimization.

## MODEL DESIGN

As depicted in Fig. 1, the article's structure begins with developing a dynamic class spacing adaptive learning model for confusion-prone classes. This algorithm dynamically adjusts class spacing and employs adaptive learning to enhance the model's generalization capability and classification accuracy within the target domain. Next, a double clustering algorithm is introduced, considering the temporal characteristics and numerical differences in auction lot prices. This approach utilizes DTW-K-medoids and FCM algorithms for precise clustering. Finally, the LSTM-based deep learning model, specifically the KF-LSTM model, integrates the results from the double clustering to forecast price confidence intervals, thereby significantly improving the accuracy of auction price predictions. These methodologies present innovative solutions for complex data classification and prediction tasks.

### Dynamic class spacing adaptive learning algorithm for confusable classes

In this adaptive learning algorithm, samples are initially randomly selected from the target domain to constitute the set Z of target domain samples in batches. These samples are then forwarded to the feature extractor to extract their features passed to the multi-classifier to minimize entropy. The loss function associated with this process is:

$$L_E = -\frac{1}{|Z|} \sum_{z \in Z} H(C_m(F(x))) \tag{1}$$

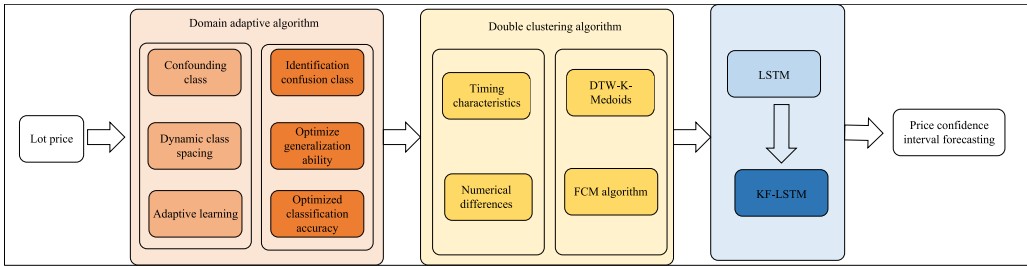

**Figure 1  Process framework.**

where $|Z|$ represents the number of samples in the target domain batch, and H($-$) denotes entropy minimization. This process aims to decrease uncertainty among the target domain samples, thereby positioning the decision boundary of unlabeled samples within regions of minimal density and enhancing sample clustering effects. Next, according to the maximum value and sub-maximum value of the output of the multi-binary classifier, where $z_i' \in Z$, $y_i$ represents the prediction of $z_i'$, $y_i \in 1, 2, \ldots, k$ represents the $y_i$ mark in the k marks, Z represents the target domain sample set in each, $P(y_i|z_i')$ is the classification probability of the target domain sample $z_i'$ in the multi-binary classifier, a and b respectively represent the category corresponding to the maximum value and sub-maximum value of the sample $z_i'$ output on the multi-binary classifier, where $a, b \in 1, 2, \ldots, k$. Since the target domain sample has no labeling information, $P(y_i|z_i')$ means that the predicted value output by the target domain sample through the classifier is a false label.

The classification distances of each target domain sample in the calculation are rearranged in the order of smallest to largest, and we select the sample with the top $l$ classification distance, *i.e.*, $d_{a,b}^1 < d_{a,b}^2 < \ldots < d_{a,b}^l$, where the classes corresponding to $d_{a,b}^i$ are $a^i$ and $b^i$, and form the set of confusable classes:

$$D_c = (a^i, b^i)_{i=1}^l. \tag{2}$$

The classification distance and boundary threshold are inversely proportional, *i.e.,* the smaller the classification distance $d_{a,b}^i$ is, the larger the boundary threshold corresponding to the confusing class $(a^i, b^i) \in D_c$ is. Next, the class spacing is dynamically adjusted and is also subject to a penalty term when it is a confusable class. A penalty is applied when $W_{y_i}^T$ and $W_j$ are feature weights for the confusion-prone class.

Finally, the features taken from the source and target domains are fed into the domain classifier D, respectively. After the adversarial training between the domain classifier D and the feature extractor F, the difference in sample distribution between the source and target domains was reduced, and domain alignment was realized. The loss function of this process is:

$$L_D = -\frac{1}{|Z_S|}\sum \log D(F(x_i)) - \frac{1}{|Z_T|}\sum \log(1 - D(F(x_j))) \tag{3}$$

where $|Z_S|$ and $|Z_T|$ are the number of samples in the source and target batch, respectively, $x_i$ is the i-th sample of the source batch, $x_j$ is the j-th sample of the target batch, $d_i$ denotes

the true domain of the i-th sample of the source batch, and $d_j$ denotes the true domain of the j-th sample of the target batch.

The total loss function is as follows:

$$L = \min_{F,C_m} \max_{D} L_E(F,C_m) + L_{DAC-CC}(F,C_m) - L_D(F,D). \tag{4}$$

## DTW- K-medoids clustering algorithm

After dynamic interval adjustment, this article constructs a DTW-based K-medoids clustering algorithm for efficiently measuring auction price curves. For any pair of $(x_i, y_i)$ within a given two time series $X = x_1, x_2, \ldots, x_m, Y = y_1, y_2, \ldots, y_n$ sequence, the distance matrix $D_{m*n}$, $(x_i, y_i)$ distance is calculated as follows

$$D(i,j) = \sqrt{(x_i - y_i)^2}. \tag{5}$$

The path from the starting point D(1,1) to the ending point D(m,n) is set as:

$$W = w_1, w_2, \ldots, w_k, \ldots, w_K, \max(m,n) \le K \le m+n-1 \tag{6}$$

Where $w_k = (k_1, k_2), D(w_k) = D(k_1, k_2)$. The path W needs to satisfy the following constraints:

Boundary conditions: $w_1 = (1,1)$, $w_k = (m,n)$

Continuity: For $w_{k-1} = (a', b'), w_k = (a, b)$, it is necessary to satisfy the $(a - a') \le 1 \cap (b - b') \le 1$

Monotonicity: For $w_{k-1} = (a', b'), w_k = (a, b), (a \ge a') \cap (b \ge b')$ needs to be satisfied.

Satisfying the above constraints and minimizing the mean of the distance values of the other passing grid yields the dynamic time-bending distance between sequence C and sequence Y, defined as:

$$DTW(X,Y) = \min_{K} \frac{1}{K} \sum_{k=1}^{K} D(w_k). \tag{7}$$

As shown in Fig. 2, we first preprocess the auction item price data, which includes data normalization and reordering by date. K initial points are then selected as potential clustering centers. We apply the dynamic DTW algorithm to calculate the distance matrix from each sample point to these K clustering centers. Based on these DTW distances, each sample point is assigned to the cluster center with the closest distance, thus forming the initial clustering structure. During the clustering process, we iteratively optimize. For each cluster, the absolute minimum error distance from all sample points within it to the cluster center is calculated, and this minimum error distance point is set as the new cluster center. This step is continuously repeated, recomputing the DTW distance matrix and updating the cluster assignments at each iteration until the centrum. Based on fuzzy set theory, data clustering is achieved by optimizing the objective function. The FCM algorithm continuously optimizes this objective function by nonlinearly minimizing a function that typically includes the Euclidean distance between cluster centroids and data points and the affiliation information of each data point to each cluster center. This approach allows

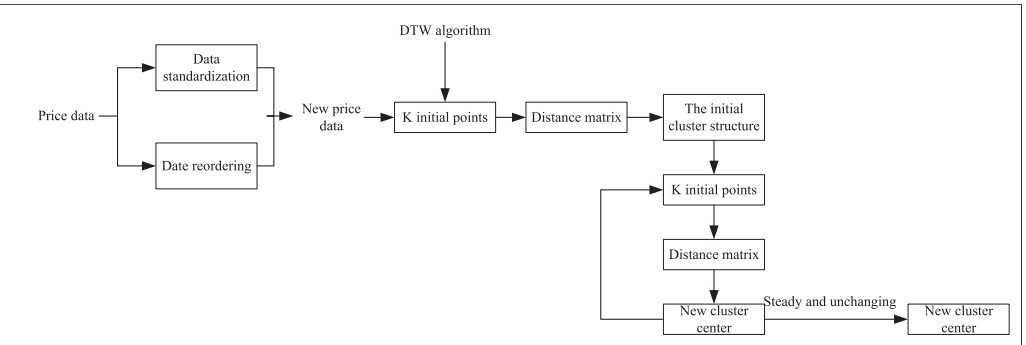

**Figure 2  K-medoids clustering algorithm based on DTW.**

FCM to deliver more flexible and accurate clustering results, particularly when handling data with ambiguity or uncertainty.

For the variable matrix $X = x_1, x_2, \ldots, x_n$, the FCM clustering algorithm aims to find a suitable degree of affiliation $U = u_{ij}$ with the clustering center $V = v_1, v_2, \ldots, v_c$ that minimizes variance and iteration error, *i.e.*, ids of all clusters stabilize and remain unchanged.

Next, we construct the FCM clustering algorithm

$$\min J(U, V) = \sum_{i=1}^{c} \sum_{j=1}^{n} u_{ij}^{*} d^{2}(x_j, v_i) \tag{8}$$

$$d_{ij} = ||x_j - v_i|| \tag{9}$$

where $J(U, V)$ is the weighted distance sum of each object in the cluster class to the clustering center, $u_{ij}^{*} \in [1, +\infty]$, indicates the degree of fuzziness of the clustering results, $d_{ij}$ indicates the Euclidean distance from the point $x_j$ to the clustering center $v_i$.

The steps of the FCM algorithm are as follows.

Step 1: Initialize the number of categories c, the iteration termination condition $\varepsilon$, and the value space of each element in the affiliation matrix $U_o$, $U_o$ is [0, 1].

Step 2: Calculate the center value of the clusters based on $U_o$ $V(k)$; Â

Step 3: Calculate the new affiliation matrix $U_o$, if $||U(k) - U(k-1)|| < \varepsilon$, then stop the loop, otherwise, set $k = k + 1$ and move to step 2.

Since auction item price data is typically time series data, clustering involves addressing two primary issues: the similarity measurement method and the selection of clustering methods. Therefore, in this section, we apply the DWT model to denoise the original power time series before clustering auction item price data, enhancing the accuracy of time series similarity calculations. The clustering algorithm in this study adopts a dual clustering approach.

In the first layer of clustering, the historical price sequence undergoes decomposition into approximate and detailed signal sequences using DWT to mitigate interference from

high-frequency fluctuations in auction item price data. Subsequently, the DTW-based K-medoids clustering algorithm is employed to morphologically cluster the extracted price principal components, forming initial clusters. In the second layer of dual clustering, the clusters from the first layer undergo further clustering using the FCM algorithm for multidimensional accurate clustering, yielding final dual clustering results. This dual clustering approach effectively leverages morphological changes and numerical distribution in auction price data.

## KF-LSTM algorithm

Each LSTM sub-network includes three crucial gate structures: the input, forget, and output. These gates are meticulously designed to manage information within the current cell state, determining whether to retain, add, or delete information. The sigmoid function operates within the range [0,1], with output values near ''0''indicating minimal information flow and values near ''1''indicating significant or complete information passage. To ensure precise control of the cell state, the LSTM model effectively utilizes these three distinct gates. Specifically, the forget gate plays a pivotal role in deciding whether to preserve or discard information from the previous cell state. The forgetting gate combines the input vector $x_t$ at the moment of $t$ with the output vector $h_{t-1}$ at the moment of t−1 through the sigmoid layer to obtain the current output vector $f_t (0 \leq f_t \leq 1)$, which is expressed by Equation (10) :

$$f_t = \sigma(W_f \cdot [h_{t-1}, x_t] + b_f). \tag{10}$$

The input gate is used to determine the new information to be added to the current cell state by performing a dot-multiplication operation between the $f_t$ vector and the $C_{t-1}$ vector, in addition to this, the new information to be saved at moment t has to pass through the sigmoid layer and the tanh layer before it can be computed Specific formulas are detailed in Equations (11) and (12).

$$i_t = \sigma[W_i \cdot [h_{t-1}, x_t] + b_i] \tag{11}$$

$$\widetilde{C}_t = \tanh[W_C \cdot [h_{t-1}, x_t] + b_C]. \tag{12}$$

The status of the old cell $C_{t-1}$ is updated to the new cell status $C_t$:

$$C_t = f_t * C_{t-1} + i_t * \widetilde{C}_t. \tag{13}$$

In the output gate, the input vector $x_t$ and the output vector $h_{t-1}$ are firstly added to the current cell state through the sigmoid layer, and finally the final output $h_t$ will be calculated through the tanh layer, and the expression of the formulae is shown in Equations (14) and (15).

$$o_t = \sigma[W_o \cdot [h_{t-1}, x_t] + b_o] \tag{14}$$

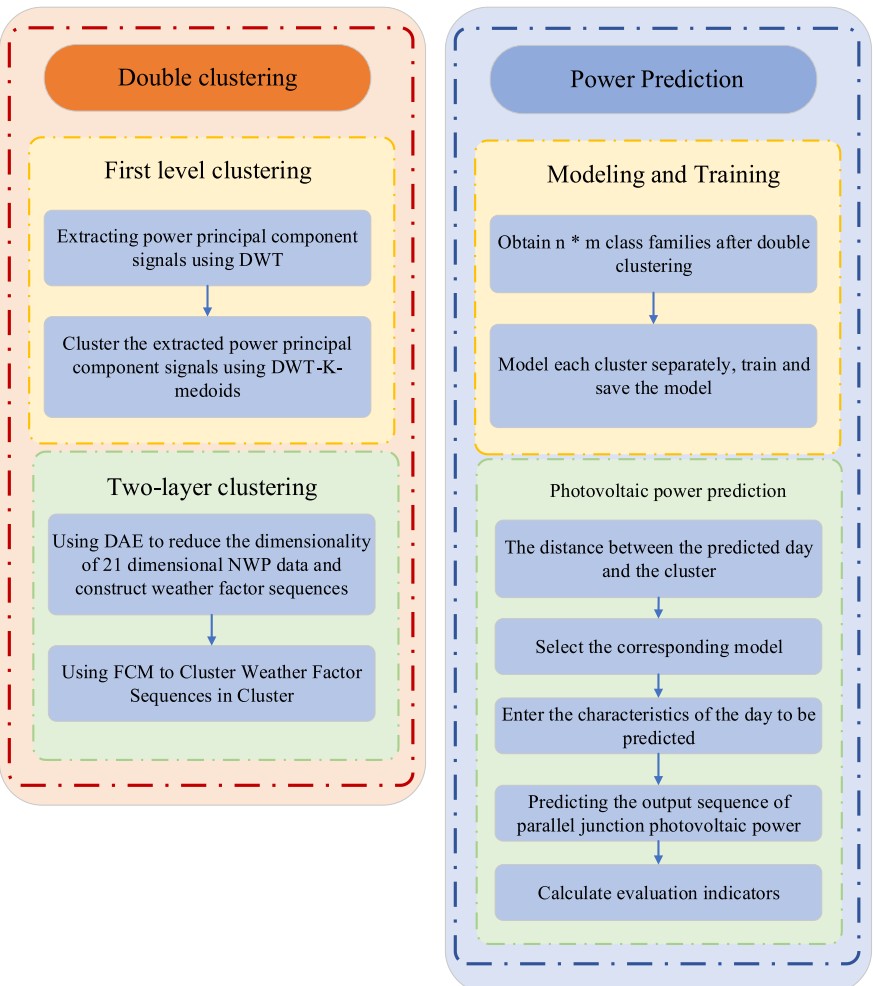

**Figure 3** **The framework of KF-LSTM.**

$$h_t = o_t * \tanh(C_t) \tag{15}$$

where W represents the weight matrix and b represents the bias matrix. Deep learning prediction is then conducted by integrating the dual clustering results, as illustrated in Fig. 3. The class clusters obtained from dual clustering are separately trained using LSTM models, and each model is saved. Subsequently, the corresponding model is selected based on the proximity between the prediction day and the class clusters. The features of the day to be predicted are then input into the selected model to predict and sequence the auction item prices accordingly.

# EXPERIMENTAL ANALYSIS

In this section, we analyze the performance of the proposed KF-LSTM model and validate its prediction accuracy using an adaptive learning algorithm by comparing it with relevant literature.

## Experimental data

To verify the accuracy of the proposed model for interval price prediction of auction items, this study utilizes data collected from the eBay online auction platform spanning from 2018 to 2022. The dataset includes auction item details such as title, description, category, auction price, auction time, and other relevant information. To ensure robustness in prediction, the collected data undergoes preprocessing steps. Duplicate entries are removed, missing values are handled, and outliers are excluded. Additionally, features are extracted from the auction time, including year, month, day of the week, and whether the day is a holiday, as these factors may influence auction item prices. Finally, 17,823 pieces of data were obtained. Then we divide them into training and test sets at an 8:2 ratio. We conducted multiple rounds of verification on the results to achieve the final average accuracy.

## Experimental evaluation criteria

In this study, three metrics, root mean square error (RMSE), mean absolute percentage error (MAPE), and accuracy rate (AR), were employed to assess the predictive performance and data characteristics of the proposed model. The formulas were computed as follows:

$$RMSE = \frac{\sqrt{\frac{1}{N}\sum_{i=1}^{N}(y-\hat{y})^2}}{P_{CAP}} \tag{16}$$

$$MAPE = \frac{\frac{1}{N}\sum_{i=1}^{N}|y-\hat{y}|}{P_{CAP}} \tag{17}$$

$$AR = 1 - \frac{\sqrt{\frac{1}{N}\sum_{i=1}^{N}(y-\hat{y})^2}}{P_{CAP}} \tag{18}$$

where N denotes the total number of samples in the test set, $y$ denotes the actual value of the auction item price, $\hat{y}$ denotes the predicted value of the auction item price, and $P_{CAP}$ denotes the total price of all auction items.

To evaluate the performance of interval price prediction, this article adopts prediction interval normalized average (PINAW) to quantify the narrowness of prediction intervals. A narrower interval width conveys more informative and practical value than a wider interval. The formula for PINAW is as follows:

$$PINAW = \frac{1}{N_t R}\sum_{i=1}^{N_t}[U_t(x_i) - L_t(x_i)] \tag{19}$$

where $N_t$ is the total number of predicted sample points, R is the difference between the predicted maximum and minimum values for data normalization, $L_t(x_i)$ is the lower limit of the predicted value, $U_t(x_i)$ is the upper limit of the predicted value, and $x_i$ is the input variable of the prediction model.

## Model comparison

Before starting the experimental analysis, let's explain the model parameter settings. The clustering setup specifies five clusters for rows and three clusters for columns, with both rows and columns utilizing the Pearson correlation coefficient as the similarity metric. This limitation prevented the algorithm from entering endless loops, ensuring that the clustering process remained efficient and practical. Additionally, we introduced a convergence criterion defined by a threshold of 0.01 on the change in clustering quality. This threshold acted as a sentinel, signaling the completion of the clustering process once the quality improvement fell below this minute margin, guaranteeing our clusters' stability and optimality.

This article introduces two comparison models for analysis alongside the double deep clustering model to demonstrate the efficacy of double clustering in the proposed method. These models include the DC(POWER-NWP)-CNN model (*Yang et al., 2022*) and the DC(DWT-NWP)-CNN model, which utilizes the DTW-based K-medoids algorithm with FCM for dual clustering (*Xian et al., 2024*).

Figures 4 and 5 depict the outcomes of dual clustering models predicting auction item prices for 2022. Figure 4 illustrates stable price changes within 40 min, showing smooth auction prices with no significant fluctuations. All three dual clustering models effectively track the actual price trends, benefiting from the strong regularity observed in auction item prices. Conversely, Fig. 5 displays fluctuating price changes within the same timeframe, where the DC(POWER-NWP)-CNN model exhibits reduced performance in tracking actual price trajectories during fluctuations. In contrast, the KF-LSTM model introduced in this study demonstrates superior prediction accuracy, particularly in capturing fluctuating moments and peak characteristics of auction prices. This effectiveness is attributed to noise reduction and dimensionality reduction techniques applied during data preprocessing, which enhance data suitability for cluster analysis. Furthermore, our dual clustering approach enhances clustering accuracy compared to traditional methods, effectively grouping samples with significant fluctuations into coherent clusters. This optimization enables individual predictors better to discern complex nonlinear relationships between inputs and outputs, thereby substantially improving overall prediction accuracy.

Figure 6 presents a comparative analysis of three different auction item price-prediction models. To comprehensively assess their performance, we meticulously selected and tested five representative sets of auction items from the test dataset. This thorough evaluation provides insights into each model's real-world performance.

As depicted in Fig. 6, the models proposed in this article demonstrate consistent prediction accuracy across challenging scenarios, including auction items with significant price fluctuations in the third dataset, despite occasional larger prediction errors. Notably, the average accuracy of our model reaches an impressive 90.23%, significantly

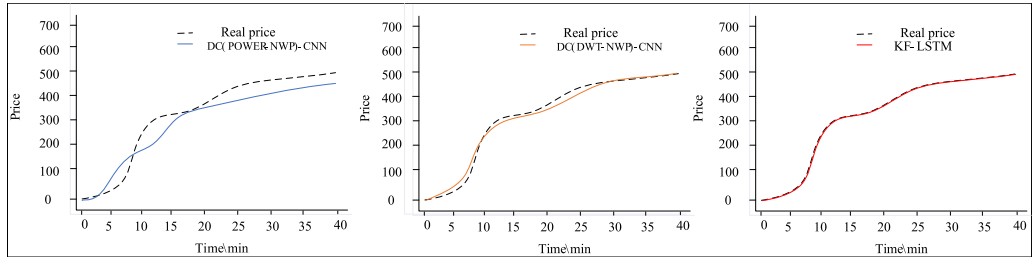

**Figure 4** Comparison of predictive performance of auction items with stable price changes.

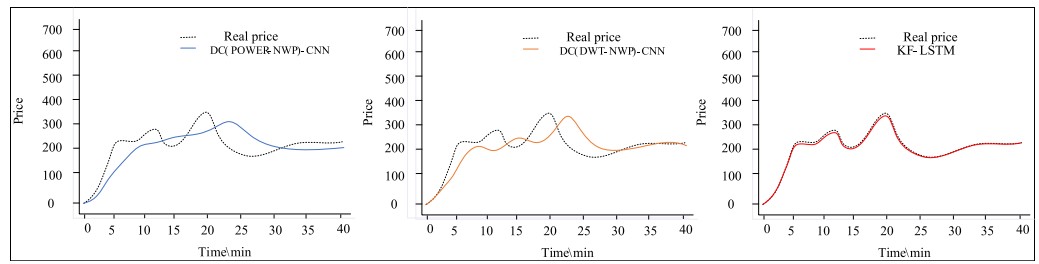

**Figure 5** Performance comparison of auction items with large price changes.

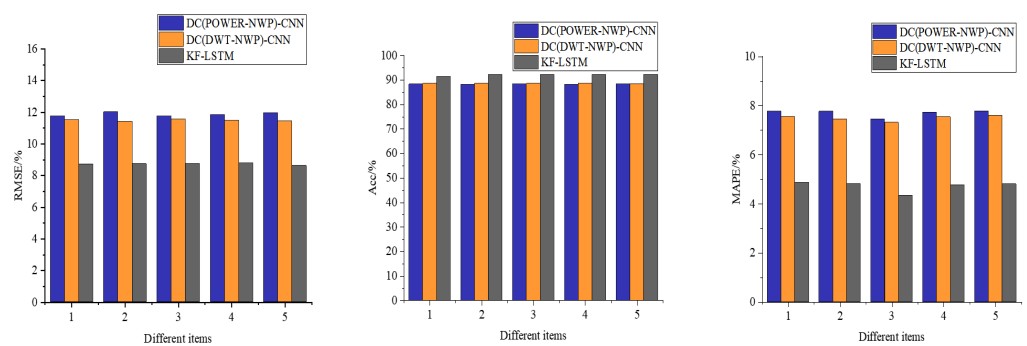

**Figure 6** Performance comparison of each model for predicting the price of auction items.

outperforming the DC(POWER-NWP)-CNN and the DC(DWT-NWP)-CNN models. Specifically, our model improves accuracy by 3.58% over the DC(POWER-NWP)-CNN model and 2.61% over the DC(DWT-NWP)-CNN model.

Furthermore, our model exhibits strong performance on MAPE, a critical indicator of prediction accuracy, with an average MAPE of only 5.41%. This figure is 2.47% and 1.89% lower than the DC(POWER-NWP)-CNN and DC(DWT-NWP)-CNN models, respectively. These outstanding results validate the effectiveness of our proposed double clustering model and underscore its practical utility in auction price prediction applications.

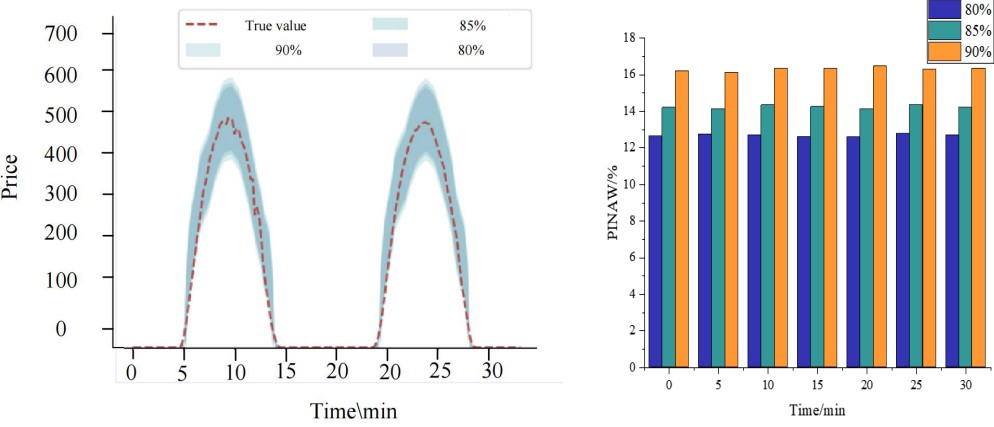

**Figure 7** **Results of interval prediction performance.**

## Interval prediction performance

Figure 7 comprehensively showcases the performance of the KF-LSTM model proposed in this article for price interval prediction. This section evaluates prediction effectiveness across different confidence levels, specifically 80%, 85%, and 90%, chosen as representative conditions for predicting auction item price intervals.

Figure 7 shows that the prediction intervals generated by the KF-LSTM model consistently encompass the actual auction item prices across all confidence levels. This highlights not only the accuracy of the model in price prediction but also underscores its reliability and practical effectiveness. Specifically, under the confidence levels of 80%, 85%, and 90%, the interval coverage of the KF-LSTM model on actual auction prices exceeds 85%. This data meets engineering requirements and demonstrates the model's robustness and consistency across varying confidence levels.

A deeper analysis reveals that the corresponding confidence interval ranges expand as confidence levels increase. This relationship aligns with statistical principles where higher confidence levels necessitate broader intervals to ensure accuracy. However, this expansion is balanced by higher interval coverage, affirming the KF-LSTM model's capability to adjust prediction strategies effectively to achieve precise auction item price interval predictions.

In summary, Fig. 7 illustrates the exceptional performance of the KF-LSTM model in price interval prediction. Regardless of the confidence level, the model consistently delivers accurate and reliable predictions, providing robust technical support for auction item price prediction applications.

## CONCLUSION

This article presents a novel dynamic class spacing adaptive learning model incorporating temporal dynamics and numerical disparities in auction item prices through the innovative KF-LSTM deep clustering approach. Our model achieves accurate price clustering and interval predictions by leveraging LSTM's strength in capturing temporal dependencies and enhancing clustering precision with a dual algorithm. The key innovation lies in its

ability to dynamically adapt to nuanced class features and effectively track price fluctuations. Future work aims to refine model performance with advanced deep learning techniques and integrate multi-source data for a comprehensive valuation and market trend analysis, thereby constructing robust and precise auction item price-prediction models.

Moreover, the model primarily relies on the temporal characteristics and numerical differences of price data, overlooking the impact of non-numerical factors such as seller reputation and item descriptions, limiting the predictions' comprehensiveness. Future research should incorporate more advanced deep learning techniques and integrate multi-source data, including unstructured information, to capture a broader range of factors influencing prices. Additionally, exploring the relationship between social and economic factors and auction prices can lead to the development of more comprehensive and accurate prediction models, thereby enhancing the adaptability and reliability of the forecasts.

### Funding
The authors received no funding for this work.

### Competing Interests
Muhammad Asif is an Academic Editor for PeerJ.

### Author Contributions

- Da Ke conceived and designed the experiments, performed the experiments, analyzed the data, prepared figures and/or tables, authored or reviewed drafts of the article, and approved the final draft.
- Xianhua Fan conceived and designed the experiments, performed the computation work, authored or reviewed drafts of the article, and approved the final draft.
- Muhammad Asif conceived and designed the experiments, performed the experiments, analyzed the data, performed the computation work, prepared figures and/or tables, authored or reviewed drafts of the article, and approved the final draft.

### Data Availability
The code is available in the Supplemental File. The Online Auctions Dataset is available at Zenodo: None. (2024). Online Auctions Dataset [Data set]. Zenodo. https://doi.org/10.5281/zenodo.11259259.

The Online Auctions Dataset was obtained from Modeling Online Auctions: https://www.modelingonlineauctions.com/datasets.

### Supplemental Information
Supplemental information for this article can be found online at http://dx.doi.org/10.7717/peerj-cs.2412#supplemental-information.

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
