# Peer review of "Adaptive learning algorithm based price prediction model for auction lots—deep clustering based interval quoting"

_PeerJ Computer Science, doi:10.7717/peerj-cs.2412_

## Round 0.1 · original submission · Major Revisions

Please see the attached comments from the reviewers. You may address all the comments and re-submit the manuscript for further considerations please.

·

Basic reporting

1. The literature review could be expanded to include studies on the application of machine learning techniques in the auction market. The paper discusses various methods, but it would be beneficial to understand how these methods have already been implemented and their effectiveness in the field.

2. Some sections could benefit from further clarification to ensure the analysis is well performed, such as the data preprocessing steps.

3. The font style is not consistent throughout the paper. It can be fixed for better readability.

3. The figures and tables could be better labeled and described. For example, figure 1 is a bit confusing since it has some text that is not mentioned in the paper.

Experimental design

1. The methodology is described in detail, but some parts need more elaboration. For example, the specific parameters and configurations used in the experiments should be clearly stated.

Validity of the findings

1. The paper should discuss the potential limitations of the proposed method and suggest areas for future research, which is missing from the paper.

·

Basic reporting

See below

Experimental design

See below

Validity of the findings

See below

Additional comments

The paper provides a clear overview of the topic— Adaptive Learning Algorithm Based Price Prediction Model for Auction Lots. It highlights the significance of constructing an auction item price prediction model based on an adaptive learning algorithm. To enhance the clarity and impact of the paper , consider the following suggestions to incorporate:

Basic:
1-Authors are required to initially use full forms. Such as Kalman Filter, Long Short-Term Memory, Double Clustering, Dynamic Time Warping etc. After using full forms authors can utilize their short forms or abbreviations.
2-The abstract should be more elaborative, including the introduction, problem statement, methodology, results, etc.
3-Explicitly state the current challenges or gaps in existing prediction models that your proposed approach aims to address.
4-There exist some grammatical errors in text i.e in line 318 the “data” is plural so there should be “undergo” instead of “undergoes”. Similarly, place “the” before model names such as in line no. 378 and 382. The authors should reconsider the text of research article.
5-Try to avoid using “To” for starting a sentence in research papers. Line 332, 342 etc.

Methodology:
6-The size of dataset is important while training the deep learning models. Therefore, authors are suggested to mention the size of dataset explicitly.
7-Elaborate briefly the appropriate reason behind using the Kalman Filter with LSTM model.
8-Discuss the parameters used in proposed model such as state transition matrix, measurement matrix in Kalman Filter and number of layers, number of units, activation function, learning rate, number of epochs in LSTM model.
9-The results were compared with (DC(POWER-NWP)-CNN and DC(DWT-NWP)-CNN), however, there is need to mention why these models were selected for comparison.

Validity of Findings:
10-Please clarify which features were the most important in predicting the auction prices.
11-Achieving 90.23 accuracy raises over-fitting concern. Authors didn’t mention any validation set for training. Therefore, model should be tested on some other dataset.
12-Please discuss the limitations of the proposed study.

Reviewer 3 ·

Basic reporting

No comments.

Experimental design

Not sufficient.

Validity of the findings

Not found significant.

Additional comments

1. The concept of Adaptive learning which the author claims in the article need to be presented in a better and formal way. It is bit confusing.
2. The equations need to be formatted properly.
3. Please follow an algorithmic approach to present all the algorithms.
4. Please format Eq. 16-18.
5. The Conclusion and Abstract section must be rewritten in a better way to show the unique outcome/contribution of the proposed models.
6. The comparative studies is not significant to claim the superiority of the proposed approach.

---

## Round 0.2 · accepted · Accept

Dear Authors, congratulations! The reviewers are satisfied with your revisions and recommended an accept decision. Please fully proofread the paper before submitting the production related copy.

·

Basic reporting

All figures are missing from the attached file. Others look good to me.

Experimental design

My previous feedback has been addressed, and everything looks good now.

Validity of the findings

My previous feedback has been addressed, and everything looks good now.

Additional comments

N/A

·

Basic reporting

Comments have been implemented. The paper is improved and accepted now.

Experimental design

Comments have been implemented. The paper is improved and accepted now.

Validity of the findings

Comments have been implemented. The paper is improved and accepted now.

Additional comments

Comments have been implemented. The paper is improved and accepted now.

Reviewer 3 ·

Basic reporting

Accepted

Experimental design

Accepted

Validity of the findings

Accepted

Additional comments

Accepted